# Early Time-Restricted Feeding Amends Circadian Clock Function and Improves Metabolic Health in Male and Female Nile Grass Rats

**DOI:** 10.3390/medicines9020015

**Published:** 2022-02-21

**Authors:** Chidambaram Ramanathan, Hayden Johnson, Suman Sharma, Wangkuk Son, Melissa Puppa, Saba Neyson Rohani, Aaryani Tipirneni-Sajja, Richard J. Bloomer, Marie van der Merwe

**Affiliations:** 1College of Health Sciences, The University of Memphis, Memphis, TN 38152, USA; rchdmbrm@memphis.edu (C.R.); ssharma9@memphis.edu (S.S.); wangkuk.s@memphis.edu (W.S.); mpuppa@memphis.edu (M.P.); rbloomer@memphis.edu (R.J.B.); 2Department of Biomedical Engineering, The University of Memphis, Memphis, TN 38152, USA; htjhnson@memphis.edu (H.J.); aaryani.sajja@memphis.edu (A.T.-S.); 3Department of Biological Sciences, The University of Memphis, Memphis, TN 38152, USA; saba.rohani@memphis.edu

**Keywords:** time-restricted feeding, obesity, metabolic syndrome, circadian rhythm, Nile grass rats

## Abstract

Lengthening the daily eating period contributes to the onset of obesity and metabolic syndrome. Dietary approaches, including energy restriction and time-restricted feeding, are promising methods to combat metabolic disorders. This study explored the effect of early and late time-restricted feeding (TRF) on weight and adiposity, food consumption, glycemic control, clock gene expression, and liver metabolite composition in diurnal Nile grass rats (NGRs). Adult male and female Nile grass rats were randomly assigned to one of three groups: (1) access to a 60% high-fat (HF) diet ad-libitum (HF-AD), (2) time-restricted access to the HF diet for the first 6 h of the 12 h light/active phase (HF-AM) or (3) the second 6 h of the 12 h light/active phase (HF-PM). Animals remained on their respective protocols for six weeks. TRF reduced total energy consumption and weight gain, and early TRF (HF-AM) reduced fasting blood glucose, restored *Per1* expression, and reduced liver lipid levels. Although sex-dependent differences were observed for fat storage and lipid composition, TRF improved metabolic parameters in both male and female NGRs. In conclusion, this study demonstrated that early TRF protocol benefits weight management, improves lipid and glycemic control, and restores clock gene expression in NGRs.

## 1. Introduction

Metabolic syndrome (MetS) is a cluster of interrelated conditions that include hypertension, dyslipidemia, and insulin resistance, and is associated with an increased risk of cardiometabolic disease. MetS is closely linked to excess adiposity, and its incidence parallels that of obesity [1]. Erratic eating behavior, such as eating multiple meals/snacks over a prolonged period each day, are commonly observed for both obesity and MetS [2,3]. This extended consumption window can result in misalignment between the rhythm of daily food intake and the circadian timing system, leading to disruption of metabolic homeostasis [4]. 

Although lifestyle modifications such as physical activity, caloric restriction, and dietary changes are effective treatments for obesity and MetS, these changes are often challenging to maintain for extended periods of time [5]. Time-restricted feeding (TRF), a form of intermittent fasting, is a dietary approach that has become popular as it has been shown to be effective in protecting against the development of obesity and MetS. TRF limits the duration of food availability over 24-h with no planned adjustment in caloric intake or food restriction [6]. This feeding protocol allows ad libitum energy intake of a person’s habitual diet within a set window of time (e.g., 3–4 h, 7–9 h, or 10–12 h), which results in an extended fasting period each day [7]. This restriction of food availability results in a reorganization of physiology and behavior directed by food-entrainable oscillators associated with the circadian clock [8].

The circadian system is a hierarchical network through which the central pacemaker, the suprachiasmatic nucleus (SCN), orchestrates temporal physiology and behavior by broadcasting rhythmic signals and systemic cues with the help of various peripheral oscillators [9]. Genetic and biochemical studies have identified clock genes and elucidated their transcriptional and post-translational feedback circuits [10]. Central to this process, the BMAL/CLOCK heterodimers act as a transcriptional activator and induce transcription of the repressor proteins PERIOD (*Per1*, *Per2*, *Per3*) and CRYPTOCHROME (*Cry1*, *Cry2*). After transcription, *Per* and *Cry* gene products inhibit their own synthesis in a negative feedback inhibition process that takes approximately 24 h [10,11]. This clock mechanism regulates the periodic expression of approximately 20% of genes in the human genome, which play an essential role in rhythmic physiological processes [12]. In rodent models, TRF can reset and synchronize the phase of these peripheral circadian oscillators, thereby shaping the pattern of whole-body metabolism, reducing weight gain, improving glycemic control, and cardiometabolic health when consuming a high-fat diet [13,14]. In humans, men with prediabetes improved insulin sensitivity, blood pressure, and oxidative stress, even in the absence of weight loss when food consumption was restricted to early in the day [7].

The most common animal models traditionally used for TRF studies are nocturnal rodent models; e.g., C57BL/6 mice. Although the circuitry of the circadian system is conserved, nocturnal animals have increased activity at night, while diurnal animals and humans increase their activity during daylight hours. The Nile grass rat (NGR) (*Arvicanthis niloticus*) has been used as a potential alternative to nocturnal rodents for traditional biomedical research [15]. Similar to humans, this rodent shows a diurnal pattern of activity/physiological processes and is often used to decipher regulatory mechanisms of circadian rhythmicity [16]. It is also an ideal animal model for the study of MetS as they spontaneously develop traits of metabolic dysfunction, including obesity, hyperglycemia, and insulin resistance [17,18]. To date, no studies have evaluated the effect of TRF of a high-fat diet on metabolic parameters in the NGR. A diurnal model would allow for the investigation of the connection between the circadian clock components and physiological responses to behavior during daylight.

This study, therefore, aimed to determine the effect of early and late TRF on high-fat diet-induced metabolic changes in male and female NGR. As the liver is the first peripheral target organ for food-induced circadian phase resetting and the main regulator of systemic metabolic homeostasis [19,20], hepatic metabolites were quantified to provide insights into the metabolic manifestations of TRF treatment [21]. 

The study demonstrates that early TRF improves body weight, metabolic parameters, restores hepatic clock gene expression, and attenuation of lipidomic effects. Although sex-dependent differences in fat partitioning and lipid composition were observed, time-restricted feeding improved metabolic parameters in both male and female Nile grass rats.

## 2. Materials and Methods

### 2.1. Experimental Animals and Feeding Protocol

All experiments were approved by the University of Memphis Institutional Animal Care and Use Committee. Nile grass rats (*Arvicanthis niloticus*) were bred and housed in a USDA-approved facility at the University of Memphis. 12–18 months male and female rats were used for the study. During the study period, animals were singly housed in Plexiglass cages in a facility where light exposure is regulated, and temperature is maintained at 22 ± 2 °C. Light exposure was structured in 12-h light-dark cycles with lights on between 8:00 and 20:00. 

Male and female rats were divided into three groups, with each group being age, weight and gender matched (*n* = 8–10 rats per group). All animals were switched to a 60% high-fat (HF) diet (Research Diets, D12492, 60% kcal fat (lard and soybean oil), 20% kcal protein, and 20% kcal carbohydrate) with a feeding schedule as follows: Group 1 (HF-AD, *n* = 10 with male:female ratio 1:1) had access to the high-fat diet ad libitum, while groups 2 and 3 were only allowed access to the food for 6 h out of every 24 h. Group 2 (HF-AM, *n* = 8 with male:female ratio 1:1) had access to food for the first 6 h of the light cycle (8:00–14:00), and group 3 (HF-PM, *n* = 10 with male:female ratio 1:1) had access to food during the second half of the light cycle (14:00–20:00). An additional age- and gender-matched group was used to demonstrate the high-fat diet effect. This group (Chow) remained on the Global Extruded Rodent Diet (Chow, 2020X Teklad, ENVIGO, Indianapolis, IN, USA) for the duration of the study and was not included in the statistical analysis. All animals had ad libitum access to water. Food consumption was monitored and recorded daily, and body mass was recorded three times a week. After five weeks on their respective protocols, body composition was determined using dual-energy X-ray absorptiometry (DEXA). All animals remained on their respective dietary protocols for 6 weeks, at which time all animals were fasted and euthanized by CO_2_ inhalation and cervical dislocations. All animals were sacrificed within a 2-h window. The tissues were harvested, weighed, immediately frozen in liquid nitrogen, and stored at −80 °C until processing.

### 2.2. Blood Parameters

After euthanasia, blood was immediately collected from the inferior vena cava into EDTA treated tubes. Blood glucose levels were immediately measured (OneTouch Ultra 2 Meter, Bayer Healthcare, Tarrytown, NY, USA). Plasma was isolated by centrifugation (2000× *g* for 15 min at 4 °C) and stored at −80 °C. Insulin (Rat/Mouse Insulin ELISA, EZRMI-13K, EMD Millipore, Billerica, MA, USA) and plasma triglycerides (Serum Triglyceride Determination Kit, Sigma TR0100, St. Louis, MO, USA) were measured as per the manufacturer’s instructions.

### 2.3. Tissue Collection and Histological Analysis

Visceral fat pads in the perigonadal region (epididymal fat pads in males and periovarian fat pads in females) and livers were harvested, weighed, and frozen in liquid nitrogen before storing them at −80 °C. Prior to freezing, a representative sample of liver tissue was also fixed in a 10% neutral buffered formalin solution (Thermo Fisher Scientific, Richard–Allan Scientific, Kalamazoo, MI-49008, USA). For histological analysis, samples were dehydrated in a graded series of ethanol solutions, cleared with Histoclear (VWR, Electron Microscopy Sciences, Hatfield, PA-19440, USA), embedded in paraffin, and sectioned into 4 µm sections. The sections were stained with hematoxylin and eosin (H&E), and analysis and documentation were performed using the EVOS imaging system.

### 2.4. Transcriptional Changes

Total RNA was isolated from frozen liver samples using Tri Reagent RT (Molecular Research Center, Cincinnati, OH, USA) for transcriptional analysis. cDNA was synthesized using a high-capacity RNA to cDNA kit (Applied Biosystems by Thermo Fisher Scientific), and quantitative PCR (qPCR) was performed using SYBR Green PCR master mix (Thermo Fisher Scientific) on a Quant Studio 6 Flex system (Applied Biosystems by Thermo Fisher Scientific). Transcript levels for each gene were normalized to *Gapdh*, and gene expression levels for *Per1* and *Per2* in the liver tissue were determined. Primers used in the qPCR analysis are listed in Appendix A.

### 2.5. Quantification of Hepatic Metabolites

About 120 mg of frozen liver tissue was homogenized in 1.74 mL methanol using a Fisherbrand bead-mill homogenizer (Thermo Fisher Scientific). Tissue homogenate equivalent to 100 mg was transferred to a centrifuge tube, and the hepatic lipids, aqueous, and protein components were separated using a methyl tert-butyl ether (MTBE) liquid-liquid extraction method [22,23]. The upper lipid (MTBE) layer and middle aqueous (MeOH/H2O) layer were removed and dried under a stream of nitrogen. Lipids were re-solubilized in 600 μL of a deuterated three solvent mixture of chloroform-d, methanol-d, and deuterium oxide (16:7:1–*v*/*v*/*v*) containing 1.18-mM dimethyl sulfone as an internal quantitative reference. Aqueous metabolites were re-solubilized in 600 μL of deuterium oxide buffered to a pH of 7.4 containing 0.16-mM 3-(Trimethylsilyl)propionic-2,2,3,3 acid as a quantitative reference.

Both lipid and aqueous spectra were obtained using a 400-MHz JEOL ECZ (Nuclear Magnetic Resonance (NMR) spectrometer. Before acquiring spectra from lipid samples, the spectrometer was cooled to 0 °C to shift the water resonance to avoid overlap with lipid peaks. For lipids, 32 FIDs were recorded with eight dummy scans using a single-pulse proton sequence with water presaturation, a pulse angle of 45°, and a relaxation delay of 4 s. Aqueous metabolite measurements were acquired using a 1D-NOESY pulse sequence at 25 °C with water presaturation, a pulse angle of 90°, and a relaxation delay of 15 s. 

The spectral processing for all samples was performed in JEOL Delta v5.3 software (Tokyo, Japan) using CRAFT [24], and the metabolite concentrations were determined from the resonance amplitudes using the following formula: $C_{x}=A_{x}*\left(\frac{C_{ref}}{A_{ref}}\right)*\left(\frac{N_{ref}}{N_{x}}\right)$, where *C_x_* is analyte concentration, *A_x_* is analyte peak area, *N_x_* is the number of protons contributing to the analyte peak, and the subscript ‘*ref*’ denotes the corresponding quantities for the quantitative reference peak in each spectrum. Chemical shifts for lipids and aqueous metabolites were determined using chemical reference standards, the human metabolome database (HMDB) [25,26], and literature values [27]. In some aqueous samples, CRAFT was not able to detect some metabolite peaks due to low signal-to-noise ratio, and these missing metabolite amplitudes were replaced using an estimated detection limit of $\frac{1}{5}$ of the minimum positive amplitude measured for the respective metabolite in all aqueous samples.

Concentrations for saturated fatty acids (SFA), unsaturated fatty acids (UFA), and polyunsaturated fatty acids (PUFA) were calculated using a previously reported method of first estimating the molar percentage of UFA-% $=\frac{\left[-CH=CH-\right]}{\left[-CH_{3}\right]}$, where the numerator corresponds to the amplitude of all olefinic acyl bonds [5.31–5.48ppm] and the denominator is the amplitude of all terminal methyl groups [0.85–1.01 ppm] [28]. The NMR signals for both monounsaturated fatty acids (MUFA) at ~2.0 ppm and total fatty acids (TFA) at ~0.88 ppm directly overlap with cholesterol proton resonances, so the unambiguous total cholesterol (TC) peak at ~0.69 ppm was used to determine the contribution of the cholesterol protons in each spectrum, and the magnitude of overlap was subtracted from the MUFA and TFA amplitudes for calculating their correct concentrations.

### 2.6. Statistical Analysis

All data are presented as mean ± SD. One-way analysis of variance (ANOVA) and Kruskal–Wallis post-hoc test were used to compare body weight change, total body fat, fasting glucose and insulin, clock gene expression, and adipose tissue and liver weight in high-fat fed animals. Two-way ANOVA was used to compare body weight and energy consumption over six weeks. Student *t*-test was used to compare the percentage of liver weight between high-fat fed males and females. The above-mentioned statistical analysis was performed with GraphPad Prism software (version 9, GraphPad, San Diego, CA, USA). ANOVA with Holm-Sidak test (Sigma Plot 14.5, Systat Software, San Jose, CA, USA) was used to compare metabolite concentrations in liver samples between high-fat diets and sexes. Metabolomic heatmaps were created for all groups using MetaboAnalyst software (https://www.metaboanalyst.ca, last accessed 29 November 2021) using auto-scaled data (mean-subtracted and divided by standard deviation). For all statistical tests, a *p*-value < 0.05 is considered as showing statistical differences.

## 3. Results

### 3.1. Early Time-Restricted High-Fat Feeding Reduced Weight Gain

To evaluate the effect of early and late TRF on weight gain when consuming a 60% high-fat diet, body weight and energy consumption were monitored throughout the study. During the first week on the TRF protocol, the HF-AM and HF-PM groups lost 10–12% (approximately 10 g) of their body mass, while the HF-AD group gained 2.9 ± 3.7 g, resulting in a significant difference between the TRF groups and the HF-AD group at the end of week 1 (Figure 1A, HF-AD vs. HF-AM, *p* = 0.003; HF-AD vs. HF-PM *p* = 0.0002). This weight loss in the TRF groups is most likely due to the acclimatization of the feeding protocol that includes extensive fasting periods. After the first week, however, both time-restricted feeding groups maintained or increased weight for the study duration. Interestingly, the body mass of the HF-AM group remained significantly different from the HF-AD group from weeks 1 to 6 (*p* = 0.02), while the HF-PM group was only significantly different until week 2 (Figure 1A, *p* = 0.03), despite no difference in food consumption between the HF-AM and HF-PM groups for the duration of the study (Figure 1B, *p* > 0.5). To assess the impact of TRF on adiposity, body composition was quantified by a dual-energy X-ray absorptiometry scan. There was a trend towards reducing fat mass between HF-AD and the HF-AM (*p* = 0.08), with no difference between the TRF groups (Figure 1C).

Although the TRF response was similar in both male and female NGRs, the ad libitum exposure of the 60% high-fat diet resulted in a sex-dependent difference in weight gain; female rats had only 1.4 ± 5.4% body mass increase, while male rats had a 7.0 ± 9.0% increase in body mass during the 6 weeks study period (Appendix A). There were no differences in food consumption between the male and female rats (data not shown).

### 3.2. Time-Restricted High Fat Feeding Altered Fasting Blood Glucose Level

Fasting blood glucose, insulin, and triglyceride were measured from blood collected at the time of sacrifice. Blood samples were also collected from a chow-fed control group as a physiological reference. Early TRF (HF-AM) significantly reduced fasting blood glucose compared to the HF-AD (Figure 2A; *p* = 0.0127). Differences seen in glucose were mirrored in the fasting insulin levels, although no significant differences were detected (Figure 2B, *p* > 0.05). No differences were detected in plasma triglycerides between the TRF groups (Figure 2C). 

### 3.3. Early TRF Improves Liver Clock Gene Expression

Previous TRF studies have shown that TRF can increase the expression of various clock components in the liver [14]. We assessed the expression of *Per1* and *Per2* at the time of sacrifice. Early TRF significantly increased *Per1* expression in liver tissue compared to HF-AD (Figure 3A; *p* = 0.04), while HF-PM did not differ from the HF-AD group (Figure 3A). Although *Per2* had a similar expression pattern as *Per1*, no significant difference was detected between the groups (Figure 3B, *p* > 0.05). Expression patterns for *Per1* and *Per1* were similar between male and female rats (Appendix A).

### 3.4. Early TRF Decreases Liver Weight and Alters Hepatic Metabolic Processes

There was a significant reduction in liver weight between the HF-AM and HF-AD groups (Figure 4A, *p* = 0.02). The reduction in the liver weight correlated with changes in hepatic lipid composition as seen in the heatmap in Figure 4B and Table 1 as determined by NMR spectroscopy. A significant difference between HF-AD and HF-AM were detected for triglycerides (TG, *p* = 0.046) and total cholesterol (TC, *p* = 0.048), and tendency towards significance was observed for total fatty acids (TFA, *p* = 0.09), saturated fatty acids (SFA, *p* = 0.07), and monounsaturated fatty acids (MUFA, *p* = 0.09). The overall hepatic lipid profile resulting from TRF, especially early TRF, reflects the composition seen in animals not exposed to a high-fat diet (Figure 4B, see Chow control group). Interestingly, phosphatidylcholine (PC) and sphingomyelin (SM) were higher in the HF-AM group and lowered in both the HF-AD and HF-PM groups (Figure 4B and Table 1). 

There was also an overall reprogramming in metabolic processes as the TRF high-fat protocol altered various water-soluble metabolites. The high-fat diet reduced almost all measured hepatic aqueous metabolites compared to a non-high-fat fed group (Figure 4C, Table 2). While the TRF protocol significantly increased levels of acetate (Table 2, *p* = 0.005), the HF-PM specifically increased alanine (*p* = 0.022) and fumarate (*p* = 0.003, Table 2).

### 3.5. Lipid Distribution Differs between Male and Female NGRs

We observed a significant difference in relative liver weight between male and female Nile grass rats when fed a high-fat diet independent from time-restricted feeding, with female livers being relatively larger (Figure 5A, *p* = 0.001). Histological analysis of the liver revealed a sex-dependent increase in lipid accumulation in the female liver tissue (Figure 5B) compared with males. Analysis of hepatic lipid species using NMR spectroscopy revealed that female rats have statistically higher levels of total triglycerides (TG), *n* − 3 fatty acids (Om3), total fatty acids (TFA), linoleic acid (LA), unsaturated fatty acids (UFA), saturated fatty acids (SFA), monounsaturated fatty acids (MUFA) and polyunsaturated fatty acids (PUFA) (Figure 5C and Table 3). Sphingomyelin was consistently higher in male liver tissue (*p* = 0.018, Figure 5C and Table 3). 

Gonadal adipose tissue weight was compared between males and females to determine the differences in fat storage. Male rats consuming a high-fat diet have significantly more adipose tissue than females (Male HD-AD vs. Female HF-AD, *p* = 0.0233). TRF reduced adipose tissue only in males, with a significant difference observed between HF-AD and HF-AM (*p* = 0.047, Figure 5D). No TRF effect was seen in adipose tissue weight in female rats (Figure 5D). 

## 4. Discussion

Obesity and MetS remain a health challenge worldwide [29]. Various dietary protocols are being implemented to prevent and treat metabolic dysfunction and related diseases [29,30]. TRF as a dietary approach became popular after it was shown that restricting the time of food consumption (limiting the number of hours you are allowed to eat per 24-h) without changing the amount or composition of the diet can improve cardiometabolic health, delay aging, and increase lifespan in rodents [14,31]. Excess calories and temporal spreading of caloric intake are the primary causes of obesity. TRF is a well-established approach in nocturnal rodent models to prevent obesity and improve metabolic processes [31]; however, little information is available on diet-induced obesity and time-restricted feeding to prevent obesity-related metabolic dysfunction in diurnal animal models. In this study, male and female Nile grass rats, a diurnal animal, were allowed access to food ad libitum or in a time-restricted manner for the first 6 h or the second 6 h of the day. Similar to what has been seen in nocturnal animals, the early TRF protocol reduced weight gain and improved metabolic function in rats, including glycemic control and fasting insulin levels.

Studies using C57BL/6 male mice demonstrated that a high-fat diet can alter the circadian locomotor activity rhythms and reduce the amplitude of clock gene expression [32,33]. The reduction in amplitude of clock mechanisms results in less robust expression of the clock and clock control genes, many of which are responsible for the metabolic processes [33]. Similar to what was seen in nocturnal rodent models, high-fat fed NGRs in the current study also had reduced expression of *Per1* concomitant with increased weight gain. Therefore, the temporal disruption in cellular metabolic processes predisposes the animal to obesity and metabolic disease. However, early TRF, where animals were fed a high-fat diet for the first 6 h of their 12-h active phase (and an 18 h fasting period), altered nutrient-entrained oscillators and metabolic regulators to revise nutrient-associated transcriptional programs that improve metabolic functions and reduce weight gain [13,14]. The results from the current study in Nile grass rats are consistent with published studies and demonstrate nutrient-entrained improvement in clock gene expression [34].

TRF altered hepatic lipids and aqueous metabolites as measured by NMR. Metabolic profiling using NMR spectroscopy is a powerful method for measuring metabolic products in response to nutrition and has been used in multiple studies to evaluate the effects of diet [35,36,37]. The reduced levels of aqueous metabolites measured in the liver of ad libitum high-fat fed group compared to the chow-fed animal are similar to a previous study in male Sprague-Dawley rats reporting lower levels of most measured amino acids as well as acetate and niacinamide in a high-fat diet versus a low-fat diet [38]. Xie et al. also demonstrated a decrease in glucogenic amino acids and an increase in hepatic lipid accumulation in response to a high-fat diet in male Wistar rats [39]. The current study shows that TRF attenuated some high-fat ad libitum diet-induced effects on aqueous metabolite levels. The two TRF groups had significantly higher hepatic acetate, while the HF-PM group had increased alanine and fumarate.

As expected, NGRs on the HF-AD diet had significantly more hepatic triglycerides, cholesterol, and fatty acids than those on the standard chow diet. Hepatic accumulation of triglycerides, cholesterol, and fatty acids are hallmarks of a high-fat diet [40,41,42]. However, lower levels of these lipids were measured in both TRF groups compared to their ad libitum counterparts, consistent with reports that TRF can induce positive lipidomic effects [43,44]. These differences were only statistically significant for the early time-restricted (HF-AM) feeding group, specifically for triglycerides and fatty acids. Hence, the lipid profiles measured in this study suggest the timing of food intake, i.e., restricting food to early in the day to be a promising preventative strategy for steatosis.

Interestingly, female rats had significantly higher hepatic triglycerides and fatty acids than males across all groups. The NMR spectroscopy results were corroborated by histological analysis of the liver, showing extensive fat accumulation in the female rats, even when consuming a low-fat diet. Male rats did not have excessive hepatic lipid accumulation but appear to divert fat storage to peripheral adipose depots. The higher concentrations of hepatic triglycerides and fatty acids and the lower amount of adipose tissue measured in females than males suggest sexual dimorphism in hepatic metabolism and fat distribution between the male and female rats profiled in this study. Similarly, Schneider et al. demonstrated sex-dependent differences in Nile grass rats with respect to cardiac function [45]. Moreover, in a nocturnal rodent model, the weight benefits associated with TRF were sex-dependent and observed in male mice only [46].

Despite differences observed between sexes, TRF and specifically early TRF resulted in improved fasting glucose levels, clock gene expression, and liver lipid accumulations in all rats. This is consistent with studies in humans suggesting that coupling food intake with the regular active period, i.e., restricting food access to early in the day, results in improved metabolic function [47,48]. A study by Dashti et al. suggests eating later in the day is associated with obesity more than eating earlier in the day [49]. Cumulatively, these reports suggest that by eating larger meals earlier in the day, the peripheral clock genes in the liver align with the central clock and generate synchronized rhythms [48].

## 5. Conclusions

In summary, the results reported here provide the first evidence that TRF has a beneficial effect on body weight and metabolic parameters in the diurnal Nile grass rats model. Early TRF improved weight management, lipid, and glycemic control and restored *Per1* expression in the liver.

A caveat of the study is that animals at sacrifice were not all fasted for the same time period. HF-AD groups were subjected to a 10-h overnight fast, while the TRF groups had different fasting times due to their respective protocols. The HF-AM group had the longest fasting time at sacrifice, specifically 18 h, while the HF-PM group fasted for 12 h. The results from this study, therefore, may not be solely related to the time of day of food consumption but also a function of the extended fasting period.

## Figures and Tables

**Figure 1 medicines-09-00015-f001:**
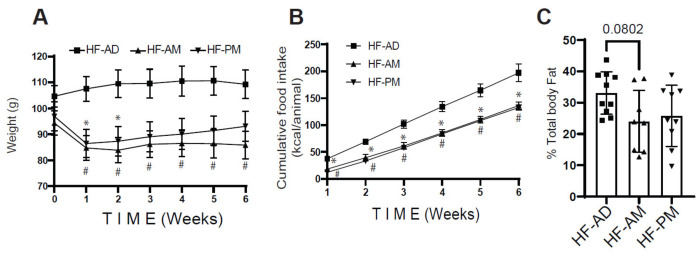
Early Time-restricted high-fat feeding reduces weight gain. Male and female Nile grass rats were fed a high fat (HF) diet (60% fat) either ad libitum (HF-AD) or in a time-restricted manner for either the first 6 h (8:00–14:00) of the active phase (HF-AM) or the second 6 h (14:00–20:00) of the active phase (HF-PM) for six weeks. Bodyweight (**A**) and energy consumption (**B**) were monitored, and body composition (**C**) was determined. Values are mean ± SD, *n* = 8–10 Nile grass rats per group. The statistical differences were determined by two-way ANOVA (**A**,**B**) or one-way ANOVA (**C**). * *p* < 0.05 between HF-AD vs. HF-PM, ^#^ *p* < 0.05 between HF-AD vs. HF-AM in (**A**,**B**). Squares and triangles in C refer to individual animal data for each group.

**Figure 2 medicines-09-00015-f002:**
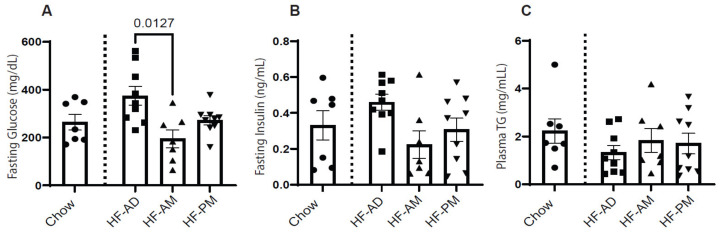
Early time-restricted high-fat feeding improves glycemic control. Blood samples were collected from all the animals at sacrifice. Glucose levels were measured using a glucometer (**A**). Plasma was harvested, and insulin (**B**) and triglyceride (**C**) levels were measured. Values are mean ± SD, *n* = 8–10 animals per group. One-way ANOVA was used to determine statistical differences, and significant values were indicated on the graph. As physiological references, data from the chow-fed animal group were included. Circles, squares, and triangles refer to individual animal data for each group.

**Figure 3 medicines-09-00015-f003:**
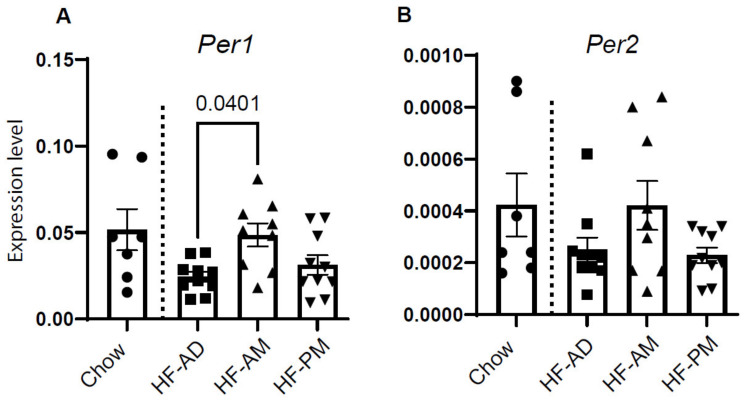
Early TRF improves clock genes expression. Expression of (**A**) *Per1* and (**B**) *Per2* were measured in liver tissue collected at the time of sacrifice. Gapdh was used as an internal control. Values are mean ± SD, *n* = 8–10 animals/group. One-way ANOVA was used to determine statistical differences, and the significant value was indicated on the graph. Circles, squares, and triangles refer to individual animal data for each group.

**Figure 4 medicines-09-00015-f004:**
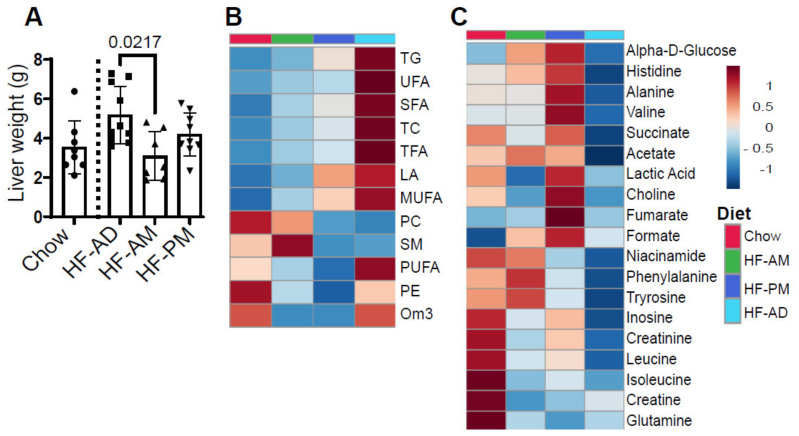
Early TRF decreases liver weight and alters hepatic lipid and aqueous metabolites. (**A**) Liver samples were collected and weighed at the time of sacrifice. Values are mean ± SD, *n* = 8–10 animals/group. The statistical differences were determined by one-way ANOVA, and the significant differences are indicated on the graph. Circles, squares, and triangles in A refer to individual animal data for each group. (**B**,**C**). Heatmap represents lipid concentrations from NMR analysis. Data were auto-scaled and clustered by average linkage using Euclidean distance. The top four colored rectangle box indicates the experimental group: Red—Chow; Green—HF-AM; Blue—HF-PM; light blue—HF-AD. Chow group data were used as a physiological reference for hepatic lipid and aqueous metabolites. The red and blue colors in the heatmap denote concentrations above and below the indicated lipid’s mean concentration value, respectively, with darker colors indicating values further from the mean. The concentration of each hepatic metabolite from figure (**B**,**C**) are shown in Table 1 and Table 2, respectively.

**Figure 5 medicines-09-00015-f005:**
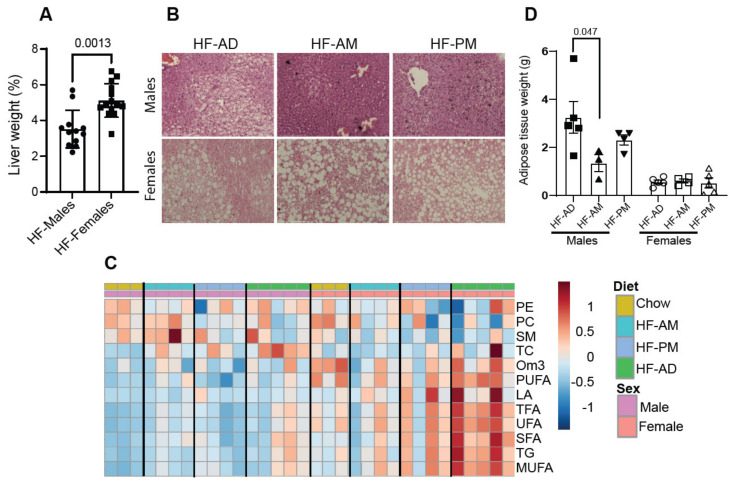
Sex-dependent lipid distribution. (**A**) Liver samples from males and females were collected and weighed. Values are mean ± SD, *n* = 12–14 animals/group. The statistical differences were determined by *t*-test, and the significant differences are indicated on the graph. (**B**) Representative hematoxylin and eosin-stained liver sections from male and female rats. Scale bars = 100 µM. (**C**) Heatmap represents lipid concentrations from NMR analysis. Data are auto-scaled and clustered by average linkage using Euclidean distance. The top four colored rectangle box indicates the experimental group: Yellow—Chow; Blue—HF-AM; Light Blue—HF-PM; Green—HF-AD. Two colored rectangle boxes below from the top box indicate the sex: Purple—males; Light Coral—females. Chow group data are used as a physiological reference for hepatic lipids. The concentration of each hepatic metabolite is shown in Table 3. (**D**) Gonadal white adipose tissue from males and females was collected and weighed at sacrifice. Values are mean ± SD, *n* = 3–5 animals/group. The statistical differences were determined by one-way ANOVA, and the significant differences are indicated. Circles, squares, and triangles in A and D refer to individual animal data for each group.

**Table 1 medicines-09-00015-t001:** Hepatic lipid compositions determined by NMR spectroscopy.

Metabolite	Chemical Shift & Multiplicity	HF-AD	HF-AM	HF-EV	*p*-Value
**PC**	3.14 (s)	2.7 ± 0.6	3.1 ± 0.4	2.7 ± 0.5	0.14
**PE**	3.08 (t)	1.6 ± 0.6	1.5 ± 0.1	1.4 ± 0.6	0.70
**SM**	3.13 (s)	0.2 ± 0.1	0.3 ± 0.1	0.2 ± 0.1	0.25
**Om3**	0.63 (s)	6.9 ± 2.4	5.1 ±2.1	5.1 ± 2.0	0.15
**TC**	4.27 (dd)	4.7 ± 2.2	2.8 ± 1.1	3.2 ± 1.2	**0.048**
**Tg**	0.92 (t)	49.6 ± 26.9	22.9 ± 13.8	31.6 ± 21.2	**0.046 ***
**TFA**	2.73 (t)	154.6 ± 65.7	98.5 ± 37.0	105.7 ± 60.7	0.09
**LA**	UFA minus MUFA	32.9 ± 33.0	16.2 ± 10.6	27.1 ± 24.7	0.39
**UFA**	0.83 (m)	96.9 ± 41.1	64.5 ± 25.9	67.1 ± 39.0	0.13
**SFA**	SFA-% times TFA	57.7 ± 26.5	34.0 ± 12.0	38.6 ± 22.1	0.07
**MUFA**	1.97 (m)	46.1 ± 22.6	25.6 ± 13.3	33.1 ± 19.1	0.09
**PUFA**	TFA minus UFA	50.8 ± 19.9	38.9 ± 14.0	34.0 ± 21.2	0.17
**UFA-%**	From equation	63% ± 6%	65% ± 2%	64% ± 4%	0.66
**SFA-%**	100 minus UFA-%	37% ± 6%	35% ± 2%	36% ± 4%	0.66
**MUFA-%**	MUFA divided by TFA	29% ± 5%	25% ± 3%	33% ± 12%	0.20
**PUFA-%**	UFA-% minus MUFA-%	34% ± 9%	40% ± 4%	31% ± 11%	0.17

Concentrations of lipids (mM) are indicated as mean ± standard deviation for all study groups. An one-way ANOVA was used to determine the diet effect on liver lipid compositions. Significant differences for are noted as follows: ***** = HF-AD higher than HF-AM. Bolded cells show significant differences induced by diet protocol. *p* = 0.04 for HF-AD vs. HF-AM. PC = phosphatidylcholine, PE = phosphatidylethanolamine, SM= Sphingomyelin, Om3 = omega-3 fatty acids, TC = total cholesterol, Tg = total triglycerides, TFA = total fatty acids, LA = Linoleic Acid, UFA = unsaturated fatty acids, SFA = saturated fatty acids, MUFA = monounsaturated fatty acids, PUFA = polyunsaturated fatty acids.

**Table 2 medicines-09-00015-t002:** Hepatic aqueous metabolites determined by NMR spectroscopy.

Metabolite	Chemical Shift & Multiplicity	HF-AD	HF-AM	HF-EV	*p*-Value
**Acetate**	1.79 (s)	0.17 ± 0.04	0.25 ± 0.05	0.24 ± 0.06	**0.005 *‡**
**Alanine**	1.35 (d)	0.42 ± 0.12	0.52 ± 0.10	0.63 ± 0.20	**0.022 ‡**
**Choline**	3.08 (s)	0.11 ± 0.05	0.11 ± 0.06	0.13 ± 0.04	0.66
**Creatine**	2.91 (s)	0.14 ± 0.11	0.09 ± 0.03	0.11 ± 0.11	0.55
**Creatinine**	2.92(s)	0.03 ± 0.02	0.03 ± 0.01	0.04 ± 0.02	0.47
**Formate**	8.33 (s)	0.07 ± 0.05	0.07 ± 0.05	0.09 ± 0.07	0.70
**Fumarate**	6.39 (s)	0.06 ± 0.03	0.06 ± 0.02	0.11 ± 0.03	**0.002 ‡‡**
**Glucose**	5.11 (d)	7.89 ± 3.30	10.58 ± 3.86	11.58 ± 3.37	0.09
**Glutamine**	2.32 (m)	1.10 ± 0.46	1.10 ± 0.36	0.97 ± 0.30	0.751
**Histidine**	7.75 (s)	0.07 ± 0.02	0.10 ± 0.03	0.11 ± 0.04	0.09
**Inosine**	8.22 (s)	0.62 ± 0.22	0.80 ± 0.27	0.88 ± 0.18	0.06
**Isoleucine**	0.89 (t)	0.08 ± 0.09	0.09 ± 0.09	0.12 ± 0.12	0.67
**Lactate**	1.20 (d)	1.56 ± 0.69	1.39 ± 0.52	2.00 ± 0.94	0.25
**Leucine**	0.83 (m)	0.10 ± 0.04	0.12 ± 0.07	0.13 ± 0.05	0.47
**Niacinamide**	8.81 (d) & 7.48 (m)	0.06 ± 0.02	0.08 ± 0.03	0.07 ± 0.02	0.08
**Phenylalanine**	7.20 (d) & 7.29 (d)	0.02 ± 0.02	0.04 ± 0.02	0.03 ± 0.02	0.14
**Succinate**	2.27 (s)	0.44 ± 0.28	0.55 ± 0.21	0.62 ± 0.16	0.28
**Tyrosine**	6.78 (d) & 7.06 (d)	0.02 ± 0.01	0.04 ± 0.02	0.03 ± 0.02	0.14
**Valine**	0.91 (d)	0.10 ± 0.02	0.14 ± 0.08	0.20 ± 0.16	0.13

Concentrations of aqueous metabolites are indicated as mean ± standard deviation for all study groups. A one-way ANOVA was used to determine the diet effect on the liver aqueous metabolites. Significant differences for each metabolite are indicated in the table. Bolded cells show significant differences induced by diet protocol. The following symbols denote significant differences: ‡ for HF-PM vs. HF-AD, ‡‡ for HF-PM vs. HF-AD and HF-AM, *‡ for HF-AM and PM vs. HF-AD.

**Table 3 medicines-09-00015-t003:** Hepatic lipid compositions from male and female NGRs determined by NMR.

Metabolite	Male			Female			Sex	Diet	Interaction
	HF-AD	HF-AM	HF-EV	HF-AD	HF-AM	HF-EV			
**PC**	2.8 ± 0.5	3.3 ± 0.3	2.8 ± 0.2	2.5 ± 0.7	3.0 ± 0.4	2.6 ± 0.8	0.26	0.16	0.94
**PE**	1.7 ± 0.3	1.5 ± 0.2	1.3 ± 0.7	1.4 ± 0.8	1.5 ± 0.1	1.4 ± 0.6	0.69	0.73	0.82
**SM**	0.3 ± 0.1	0.4 ± 0.1	0.3 ± 0.1	0.2 ± 0.1	0.2 ± 0.05	0.2 ± 0.1	**0.018 ^m^**	0.20	0.53
**Om3**	5.7 ± 2.0	4.8 ± 2.0	4.2 ± 1.9	8.2 ± 2.3	5.5 ± 2.5	6.1 ± 1.7	**0.049 ^f^**	0.13	0.64
**TC**	5.1 ± 1.5	3.6 ± 1.0	3.5 ± 1.5	4.3 ± 2.9	2.1 ± 0.5	2.8 ± 0.9	0.14	0.05	0.89
**Tg**	30.6 ± 15.8	17.9 ± 5.9	17.7 ± 9.9	68.6 ± 21.8	27.9 ± 18.5	45.6 ± 20.8	**0.001 ^f^**	**0.009 #**	0.59
**TFA**	103.3 ± 37.5	90.0 ± 19.9	62.9 ± 28.1	205.8 ± 41.7	107.1 ± 51.1	148.5 ± 54.1	**<0.001 ^f^**	**0.015 #***	0.09
**LA**	14.3 ± 5.7	10.3 ± 6.6	15.6 ± 12.1	51.5 ± 39.5	22.1 ± 11.3	38.5 ± 30.4	**0.014 ^f^**	0.31	0.50
**UFA**	63.7 ± 20.1	55.5 ± 13.6	38.9 ± 13.9	130.1 ± 25.4	73.5 ± 34.2	95.3 ± 35.2	**<0.001 ^f^**	**0.020 #***	0.14
**SFA**	39.6 ± 18.5	34.4 ± 7.1	24.0 ± 14.3	75.7 ± 20.6	33.6 ± 16.9	53.1 ± 19.1	**0.004 ^f^**	**0.017 #**	0.08
**MUFA**	28.4 ± 14.4	19.6 ± 6.2	20.7 ± 10.2	63.8 ± 12.7	31.6 ± 16.7	45.6 ± 18.3	**<0.001 ^f^**	**0.015 #**	0.22
**PUFA**	35.3 ± 9.0	35.9 ± 11.1	18.2 ± 10.1	66.3 ± 14.5	41.9 ± 17.6	49.7 ± 17.0	**<0.001 ^f^**	**0.041 ***	0.11
**UFA-%**	63% ± 7%	62% ± 3%	64% ± 6%	63% ± 5%	69% ± 2%	64% ± 2%	0.18	0.64	0.23
**SFA-%**	37% ± 7%	38% ± 3%	36% ± 6%	37% ± 5%	31% ± 2%	36% ± 2%	0.18	0.64	0.23
**MUFA-%**	26% ± 6%	22% ± 6%	35% ± 17%	31% ± 1%	29% ± 3%	30% ± 3%	0.48	0.20	0.30
**PUFA-%**	37% ± 12%	39% ± 4%	29% ± 17%	32% ± 5%	40% ± 4%	34% ± 2%	0.91	0.20	0.58

Concentrations of lipid (mM) are indicated as mean ± standard deviation for all study groups. A two-way ANOVA was used to determine sex, diet, and interaction effects on the liver lipid compositions. The significant differences for each lipid are indicated in the table. Bolded cells show significant differences in sex and diets. The following symbols denote sex and dietary differences: # HF-AD vs. HF-AM, * HF-AD vs. HF-PM, #* HF-AD vs. HF-AM and HF-PM, “^f^”—females > males, and “^m^”—for males > females. PC = phosphatidylcholine, PE = phosphatidylethanolamine, SM = Sphingomyelin, Om3 = omega-3 fatty acids, TC = total cholesterol, Tg = total triglycerides, TFA = total fatty acids, LA = Linoleic Acid, UFA = unsaturated fatty acids, SFA = saturated fatty acids, MUFA = monounsaturated fatty acids, PUFA = polyunsaturated fatty acids.

## Data Availability

Data supporting the reported results were generated during the study and are not publicly available. Summary of the results related to this study can be accommodated on request from the corresponding author.

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
