# Peer review of "Early Time-Restricted Feeding Amends Circadian Clock Function and Improves Metabolic Health in Male and Female Nile Grass Rats"

_medicines, 2022, doi:10.3390/medicines9020015_

Round 1

Reviewer 1 Report

This is quite an interesting and original paper. 

In the present study, Ramanathan et al. assessed the difference between early and late time-limited feeding on alteration of the circadian clock and metabolic health in Nile grass rats. The obesity epidemic occurring today forces us to search for new nutritional strategies, and TRE seems to be an effective method in humans and animals. The author's comparison of early and late TRE is interesting and crucial. And although intermittent fasting has a lot of scientific attention thru the last decade, the comparison of the different timing of TRF seems to be still original. The essential element of the present study is the animals used in the experiment. Most studies of TRF use nocturnal rodents. The Nile grass rat rodent shows a diurnal pattern of activity and can be an important tool to understand humans' mechanisms of circadian rhythmicity.

The paper is well written, and the text is clear. The conclusions are consistent with the evidence and address the main question.

Author Response

NO comments from reviewer 1

Reviewer 2 Report

Overall the study is good, but the novelty is questionable

Why is it important to repeat the experiment in NGRs when studies in humans and other animals already report what you found? I think there should be a justification for this in the manuscript.

The research design should mention how many (out of 8-10 per group) males and females were in each group, especially when you talk about sex-dependent differences. Was that male: female ratio and numbers enough to statistically conclude that the difference actually means something. What was the sample size and power for this analysis?

I think having two groups with the two time-restriction regimens and a normal diet would have been good as a comparison.  

Just like you did for the glucose, insulin, and TGs, I think it would be best to also present the data was the rats that were on the normal diet for all other parameters.

"reduces weight gain" is different from "weight loss". There seems to be confusion in the text and the figure. 

Also, please specify where there is "no difference" among groups and where there is "no difference" from the baseline. Baseline data seems to be missing.

Why did the cumulative food intake increase over time? How is that correlated to weight loss?

Some methods are described in unnecessary detail. You may not want to repeat the full protocol if it is a standard procedure. Only mention variations if any. 

Author Response

Comment 1: Why is it important to repeat the experiment in NGRs when studies in humans and other animals already report what you found? I think there should be a justification for this in the manuscript.

Response: We appreciate the reviewer's comment. Both the nocturnal animal model and humans demonstrate a similar effect of time-restricted feeding. However, to examine the connection between the circadian clock components and behavior, it is critical to identify a model where behavior mimics that seen in human populations (except in the case of night-shift workers). The diurnal grass rat is a well-characterized rodent model frequently used in circadian rhythm research. Therefore, our study aims to demonstrate the validity of this model in feeding studies, specifically as it pertains to metabolic syndrome. We have added this justification to the introduction. Please see lines 66 to 78.  

Comment 2: The research design should mention how many (out of 8-10 per group) males and females were in each group, especially when you talk about sex-dependent differences. Was that male: female ratio and numbers enough to statistically conclude that the difference actually means something. What was the sample size and power for this analysis?

Response: We thank the reviewer for this critical observation. The groups had the following number of males and females in each group: HF-AD (5 males, 5 females), HF- AM (4 males, 4 females), HF-PM (5 males, 5 females). Therefore a ratio of 1:1 males to females for all groups. The number of animals per group used is similar to our previous work in mice. For metabolic parameters and clock gene expression, male and female animals had a similar response to TRF, and therefore the numbers are sufficient to determine statistical significance. The unexpected finding related to the sex-dependent differences has limited numbers. However, some effect is so consistent, for example, triglycerides concentration in the liver (Table 3) shows statistical difference. We updated the information in the method section; please see lines  98 to 104.

Comment 3: I think having two groups with the two time-restriction regimens and a normal diet would have been good as a comparison.

Responses: We thank the reviewer for this comment. However, using a control diet with a different nutrient composition adds an additional variable to the experiment. Many journals that publish nutrition studies do not accept the “Chow diet” as a control. For this experiment, we wanted to answer whether eating food with identical composition but where access is restricted will alter the physiological outcome to the dietary protocol vs. ad libitum access. We have included the chow group to show what a “physiologically healthy” animal will look like, but this group was not included in the statistical analysis. This is now clarified in the methods (Lines 106-108).

Comment 4:Just like you did for the glucose, insulin, and TGs, I think it would be best to also present the data was the rats that were on the normal diet for all other parameters.

Responses: We agreed with this observation and included the “chow group” in Figures 3 and 4A to demonstrate what clock gene expression and liver size would look like in a “healthy” mouse.  Please see the updated figures 2 and 4.

Comment 4:

"reduces weight gain" is different from "weight loss". There seems to be confusion in the text and the figure.

Response: The “weight loss” in lines 204-205 refers to the weight loss during the first week of the experiment, where the fasted rats lost about 12% of their body mass. The title of the figure legend (Figure 1; lines 223 to 224) refers to the experiment's outcome where we observed that after the initial weight loss, the TRF-AM group gained weight slower than the TRF-PM group over the next 5 weeks. There is still a statistically significant difference between HF-AD vs. HF-AM but is not seen between HF-AF vs. HF-PM.

Comment 5: Also, please specify where there is "no difference" among groups and where there is "no difference" from the baseline. Baseline data seems to be missing.

Response: We thank the reviewer for the comment. All groups were matched for weight, age, and sex at the start of the experiment, and all analyses were performed between the groups. We did not analyze the endpoint vs. baseline within the group.

 Comment 6: Why did the cumulative food intake increase over time? How is that correlated to weight loss?.

Response: Thank you for the comment. Cumulative food intake refers to the amount of food consumed over time. As time increases, food consumption increases unless no food is consumed. This presentation of cumulative food intake is well established in the field of time-restricted feeding. In the manuscript, we did not correlate food intake to weight loss. However, we did make the point that despite consuming the exact same amount of food over the 6 weeks-as seen in figure 1B, the TRF-AM group had reduced/slower weight gain compared to the TRF-PM group, as seen in Figure 1A.

Comment 7: Some methods are described in unnecessary detail. You may not want to repeat the full protocol if it is a standard procedure. Only mention variations if any

Response: We thank the reviewer for the comment. We believe method sections 2.1-2.4 listed only the standard information. Section 2.5 (NMR Sample Preparation and Acquisition) is a novel method for lipid analysis using NMR and is described in more detail.

Round 2

Reviewer 2 Report

Revisions are satisfactory